# D-VAE: A Variational Autoencoder for Directed Acyclic Graphs

**Muhan Zhang, Shali Jiang, Zhicheng Cui, Roman Garnett, Yixin Chen**
Department of Computer Science and Engineering
Washington University in St. Louis
{muhan, jiang.s, z.cui, garnett}@wustl.edu, chen@cse.wustl.edu

## Abstract

Graph structured data are abundant in the real world. Among different graph types, directed acyclic graphs (DAGs) are of particular interest to machine learning researchers, as many machine learning models are realized as computations on DAGs, including neural networks and Bayesian networks. In this paper, we study deep generative models for DAGs, and propose a novel DAG variational autoencoder (D-VAE). To encode DAGs into the latent space, we leverage graph neural networks. We propose an asynchronous message passing scheme that allows encoding the computations on DAGs, rather than using existing simultaneous message passing schemes to encode local graph structures. We demonstrate the effectiveness of our proposed D-VAE through two tasks: neural architecture search and Bayesian network structure learning. Experiments show that our model not only generates novel and valid DAGs, but also produces a smooth latent space that facilitates searching for DAGs with better performance through Bayesian optimization.

## 1 Introduction

Many real-world problems can be posed as optimizing of a directed acyclic graph (DAG) representing some computational task. For example, the architecture of a neural network is a DAG. The problem of searching optimal neural architectures is essentially a DAG optimization task. Similarly, one critical problem in learning graphical models – optimizing the connection structures of Bayesian networks [1], is also a DAG optimization task. DAG optimization is pervasive in other fields as well. In electronic circuit design, engineers need to optimize DAG circuit blocks not only to realize target functions, but also to meet specifications such as power usage and operating temperature.

DAG optimization is a hard problem. Firstly, the evaluation of a DAG's performance is often time-consuming (e.g., training a neural network). Secondly, state-of-the-art black-box optimization techniques such as simulated annealing and Bayesian optimization primarily operate in a continuous space, thus are not directly applicable to DAG optimization due to the discrete nature of DAGs. In particular, to make Bayesian optimization work for discrete structures, we need a kernel to measure the similarity between discrete structures as well as a method to explore the design space and extrapolate to new points. Principled solutions to these problems are still lacking.

Is there a way to circumvent the trouble from discreteness? The answer is yes. If we can **embed all DAGs to a continuous space** and make the space relatively smooth, we might be able to directly use principled black-box optimization algorithms to optimize DAGs in this space, or even use gradient methods if gradients are available. Recently, there has been increased interest in training generative models for discrete data types such as molecules [2, 3], arithmetic expressions [4], source code [5], undirected graphs [6], etc. In particular, Kusner et al. [3] developed a grammar variational autoencoder (G-VAE) for molecules, which is able to encode and decode molecules into and from a **continuous latent space**, allowing one to optimize molecule properties by searching in this well-

behaved space instead of a discrete space. Inspired by this work, we propose to also train a variational autoencoder for DAGs, and optimize DAG structures in the latent space via Bayesian optimization.

To encode DAGs, we leverage graph neural networks (GNNs) [7]. Traditionally, a GNN treats all nodes symmetrically, and extracts local features around nodes by **simultaneously** passing all nodes' neighbors' messages to themselves. However, such a simultaneous message passing scheme is designed to learn local structure features. It might not be suitable for DAGs, since in a DAG: 1) nodes are not symmetric, but intrinsically have some ordering based on its dependency structure; and 2) we are more concerned about the computation represented by the entire graph, not the local structures.

In this paper, we propose an **asynchronous message passing scheme** to encode the computations on DAGs. The message passing no longer happens at all nodes simultaneously, but respects the computation dependencies (the partial order) among the nodes. For example, suppose node A has two predecessors, B and C, in a DAG. Our scheme does not perform feature learning for A until the feature learning on B and C are both finished. Then, the aggregated message from B and C is passed to A to trigger A's feature learning. This means, although the message passing is not simultaneous, it is also not completely unordered – some synchronization is still required. We incorporate this feature learning scheme in both our encoder and decoder, and propose DAG *variational autoencoder* (D-VAE). D-VAE has an excellent theoretical property for modeling DAGs– we prove that D-VAE can **injectively** encode **computations** on DAGs. This means, we can build a mapping from the discrete space to a continuous latent space so that **every** DAG computation has its **unique** embedding in the latent space, which **justifies** performing optimization in the latent space instead of the original design space.

Our contributions in this paper are: 1) We propose D-VAE, a variational autoencoder for DAGs using a novel asynchronous message passing scheme, which is able to injectively encode computations. 2) Based on D-VAE, we propose a new DAG optimization framework which performs Bayesian optimization in a continuous latent space. 3) We apply D-VAE to two problems, neural architecture search and Bayesian network structure learning. Experiments show that D-VAE not only generates novel and valid DAGs, but also learns smooth latent spaces effective for optimizing DAG structures.

## 2 Related work

**Variational autoencoder (VAE)** [8, 9] provides a framework to learn both a probabilistic generative model $p_\theta(\mathbf{x}|\mathbf{z})$ (the decoder) as well as an approximated posterior distribution $q_\phi(\mathbf{z}|\mathbf{x})$ (the encoder). VAE is trained through maximizing the evidence lower bound

$$\mathcal{L}(\phi, \theta; \mathbf{x}) = \mathbb{E}_{\mathbf{z} \sim q_\phi(\mathbf{z}|\mathbf{x})}[\log p_\theta(\mathbf{x}|\mathbf{z})] - \mathrm{KL}[q_\phi(\mathbf{z}|\mathbf{x}) \| p(\mathbf{z})]. \tag{1}$$

The posterior approximation $q_\phi(\mathbf{z}|\mathbf{x})$ and the generative model $p_\theta(\mathbf{x}|\mathbf{z})$ can in principle take arbitrary parametric forms whose parameters $\phi$ and $\theta$ are output by the encoder and decoder networks. After learning $p_\theta(\mathbf{x}|\mathbf{z})$, we can generate new data by decoding latent space vectors $\mathbf{z}$ sampled from the prior $p(\mathbf{z})$. For generating discrete data, $p_\theta(\mathbf{x}|\mathbf{z})$ is often decomposed into a series of decision steps.

**Deep graph generative models** use neural networks to learn distributions over graphs. There are mainly three types: token-based, adjacency-matrix-based, and graph-based. Token-based models [2, 3, 10] represent a graph as a sequence of tokens (e.g., characters, grammar rules) and model these sequences using RNNs. They are less general since task-specific graph grammars such as SMILES for molecules [11] are required. Adjacency-matrix-based models [12, 13, 14, 15, 16] leverage the proxy adjacency matrix representation of a graph, and generate the matrix in one shot or generate the columns/entries sequentially. In contrast, graph-based models [6, 17, 18, 19] seem more natural, since they operate directly on graph structures (instead of proxy matrix representations) by iteratively adding new nodes/edges to a graph based on the existing graph and node states. In addition, the graph and node states are learned by **graph neural networks (GNNs)**, which have already shown their powerful graph representation learning ability on various tasks [20, 21, 22, 23, 24, 25, 26, 27].

**Neural architecture search (NAS)** aims at automating the design of neural network architectures. It has seen major advances in recent years [28, 29, 30, 31, 32, 33]. See Hutter et al. [34] for an overview. NAS methods can be mainly categorized into: 1) reinforcement learning methods [28, 31, 33] which train controllers to generate architectures with high rewards in terms of validation accuracy, 2) Bayesian optimization based methods [35] which define kernels to measure architecture similarity and extrapolate the architecture space heuristically, 3) evolutionary approaches [29, 36, 37] which use evolutionary algorithms to optimize neural architectures, and 4) differentiable methods

[32, 38, 39] which use continuous relaxation/mapping of neural architectures to enable gradient-based optimization. In Appendix A, we include more detailed discussion on several most related works.

**Bayesian network structure learning (BNSL)** is to learn the structure of the underlying Bayesian network from observed data [40, 41, 42, 43]. Bayesian network is a probabilistic graphical model encoding conditional dependencies among variables via a DAG [1]. One main approach for BNSL is score-based search, i.e., define some "goodness-of-fit" score for network structures, and search for one with the optimal score in the discrete design space. Commonly used scores include BIC and BDeu, mostly based on marginal likelihood [1]. Due to the NP-hardness [44], however, exact algorithms such as dynamic programming [45] or shortest path approaches [46, 47] can only solve small-scale problems. Thus, people have to resort to heuristic methods such as local search and simulated annealing, etc. [48]. BNSL is still an active research area [41, 43, 49, 50, 51].

# 3   DAG variational autoencoder (D-VAE)

In this section, we describe our proposed DAG variational autoencoder (D-VAE). D-VAE uses an asynchronous message passing scheme to encode and decode DAGs. In contrast to the simultaneous message passing in traditional GNNs, D-VAE allows encoding *computations* rather than *structures*.

**Definition 1.** *(Computation) Given a set of elementary operations $\mathcal{O}$, a computation $C$ is the composition of a finite number of operations $o \in \mathcal{O}$ applied to an input signal $x$, with the output of each operation being the input to its succeeding operations.*

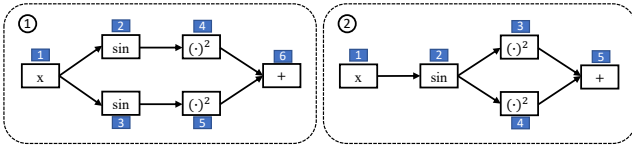

Figure 1: Computations can be represented by DAGs. Note that the left and right DAGs represent the same computation.

The set of elementary operations $\mathcal{O}$ depends on specific applications. For example, when we are interested in computations given by a calculator, $\mathcal{O}$ will be the set of all the operations defined on the functional buttons, such as $+$, $-$, $\times$, $\div$, etc. When modeling neural networks, $\mathcal{O}$ can be a prede-fined set of basic layers, such as $3{\times}3$ convolution, $5{\times}5$ convolution, $2{\times}2$ max pooling, etc. A computation can be represented as a directed acyclic graph (DAG), with directed edges representing signal flow directions among node operations. The graph must be acyclic, since otherwise the input signal will go through an infinite number of operations so that the computation never stops. Figure 1 shows two examples. Note that the two different DAGs in Figure 1 represent the same computation, as the input signal goes through exactly the same operations. We discuss it further in Appendix B.

## 3.1   Encoding

We first introduce the encoder of D-VAE, which can be seen as a graph neural network (GNN) using an asynchronous message passing scheme. Given a DAG $G$, we assume there is a single starting node which does not have any predecessors (e.g., the input layer of a neural architecture). If there are multiple such nodes, we add a virtual starting node connecting to all of them.

Similar to standard GNNs, we use an update function $\mathcal{U}$ to compute the hidden state of each node based on its neighbors' incoming message. The hidden state of node $v$ is given by:

$$\mathbf{h}_v = \mathcal{U}(\mathbf{x}_v, \mathbf{h}_v^{\text{in}}), \tag{2}$$

where $\mathbf{x}_v$ is the one-hot encoding of $v$'s type, and $\mathbf{h}_v^{\text{in}}$ represents the incoming message to $v$. $\mathbf{h}_v^{\text{in}}$ is given by aggregating the hidden states of $v$'s predecessors using an aggregation function $\mathcal{A}$:

$$\mathbf{h}_v^{\text{in}} = \mathcal{A}(\{\mathbf{h}_u : u \to v\}), \tag{3}$$

where $u \to v$ denotes there is a directed edge from $u$ to $v$, and $\{\mathbf{h}_u : u \to v\}$ represents a multiset of $v$'s predecessors' hidden states. If an empty set is input to $\mathcal{A}$ (corresponding to the case for the starting node without any predecessors), we let $\mathcal{A}$ output an all-zero vector.

Compared to the traditional simultaneous message passing, in D-VAE the message passing for a node must wait until all of its predecessors' hidden states have already been computed. This simulates how a computation is really performed – to execute some operation, we also need to wait until all its input signals are ready. So how to make sure all the predecessor states are available when a new node comes? One solution is that we can sequentially perform message passing for nodes following a *topological ordering* of the DAG. We illustrate this encoding process in Figure 2.

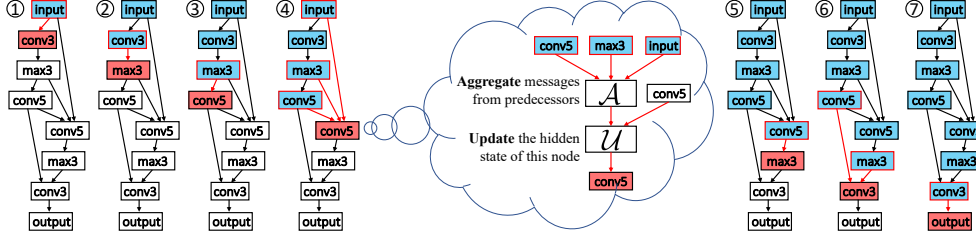

Figure 2: An illustration of the encoding procedure for a neural architecture. Following a topological ordering, we iteratively compute the hidden state for each node (red) by feeding in its predecessors' hidden states (blue). This simulates how an input signal goes through a computation, with $\mathbf{h}_v$ simulating the output signal at node $v$.

After all nodes' hidden states are computed, we use $\mathbf{h}_{v_n}$, the hidden state of the ending node $v_n$ without any successors, as the output of the encoder. Then we feed $\mathbf{h}_{v_n}$ to two MLPs to get the mean and variance parameters of the posterior approximation $q_\phi(\mathbf{z}|G)$ in (1). If there are multiple nodes without successors, we again add a virtual ending node connecting from all of them.

Note that although topological orderings are usually not unique for a DAG, we can take any one of them as the message passing order while ensuring the encoder output is always the same, revealed by the following theorem. We include all theorem proofs in the appendix.

**Theorem 1.** *The* D-VAE *encoder is invariant to node permutations of the input* DAG *if the aggregation function $\mathcal{A}$ is invariant to the order of its inputs.*

Theorem 1 means isomorphic DAGs are always encoded the same, no matter how we index the nodes. It also indicates that so long as we encode a DAG complying with its partial order, we can perform message passing in arbitrary order (even parallelly for some nodes) with the same encoding result.

The next theorem shows another property of D-VAE that is crucial for its success in modeling DAGs, i.e., it is able to injectively encode computations on DAGs.

**Theorem 2.** *Let $G$ be any* DAG *representing some computation $C$. Let $v_1, \ldots, v_n$ be its nodes following a topological order each representing some operation $o_i, 1 \leq i \leq n$, where $v_n$ is the ending node. Then, the encoder of* D-VAE *maps $C$ to $\mathbf{h}_{v_n}$ injectively if $\mathcal{A}$ is injective and $\mathcal{U}$ is injective.*

The significance of Theorem 2 is that it provides a way to injectively encode computations on DAGs, so that every computation has a unique embedding in the latent space. Therefore, instead of performing optimization in the original discrete space, we may alternatively perform optimization in the **continuous latent space**. In this well-behaved Euclidean space, distance is well defined, and principled Bayesian optimization can be applied to search for latent points with high performance scores, which transforms the discrete optimization problem into an easier continuous problem.

Note that Theorem 2 states D-VAE injectively encodes computations on graph structures, rather than graph structures themselves. Being able to injectively encode graph structures is a very strong condition, as it implies an efficient algorithm to solve the challenging graph isomorphism (GI) problem. Luckily, here what we really care about are computations instead of structures, since we do not want to differentiate two different structures $G_1$ and $G_2$ as long as they represent the **same computation**. Figure 1 shows such an example. Our D-VAE can identify that the two DAGs in Figure 1 actually represent the same computation by encoding them to the same vector, while those encoders focusing on encoding structures might fail to capture the underlying computation and output different vectors. We discuss more advantages of Theorem 2 in optimizing DAGs in Appendix G.

To model and learn the injective functions $\mathcal{A}$ and $\mathcal{U}$, we resort to neural networks thanks to the universal approximation theorem [52]. For example, we can let $\mathcal{A}$ be a gated sum:
$$\mathbf{h}_v^{\text{in}} = \sum_{u \to v} g(\mathbf{h}_u) \odot m(\mathbf{h}_u), \tag{4}$$
where $m$ is a mapping network and $g$ is a gating network. Such a gated sum can model injective multiset functions [53], and is invariant to input order. To model the injective update function $\mathcal{U}$, we can use a gated recurrent unit (GRU) [54], with $\mathbf{h}_v^{\text{in}}$ treated as the input hidden state:
$$\mathbf{h}_v = \text{GRU}_e(\mathbf{x}_v, \mathbf{h}_v^{\text{in}}). \tag{5}$$
Here the subscript $e$ denotes "encoding". Using a GRU also allows reducing our framework to traditional sequence to sequence modeling frameworks [55], as discussed in 3.4.

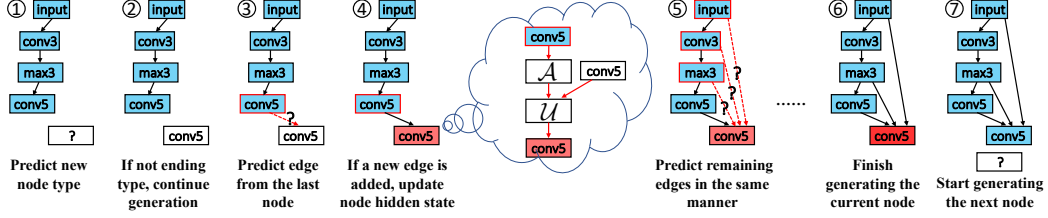

Figure 3: An illustration of the steps for generating a new node.

The above aggregation and update functions can be used to encode general computation graphs. For neural architectures, depending on how the outputs of multiple previous layers are aggregated as the input to a next layer, we will make a modification to (4), which is discussed in Appendix E. For Bayesian networks, we also make some modifications to their encoding due to the special d-separation properties of Bayesian networks, which is discussed in Appendix F.

## 3.2 Decoding

We now describe how D-VAE decodes latent vectors to DAGs (the generative part). The D-VAE decoder uses the same asynchronous message passing scheme as in the encoder to learn intermediate node and graph states. Similar to (5), the decoder uses another GRU, denoted by $\mathrm{GRU}_d$, to update node hidden states during the generation. Given the latent vector $\mathbf{z}$ to decode, we first use an MLP to map $\mathbf{z}$ to $\mathbf{h_0}$ as the initial hidden state to be fed to $\mathrm{GRU}_d$. Then, the decoder constructs a DAG node by node. For the $i^{\text{th}}$ generated node $v_i$, the following steps are performed:

1. Compute $v_i$'s type distribution using an MLP $f_{\text{add\_vertex}}$ (followed by a softmax) based on the current graph state $\mathbf{h}_G := \mathbf{h}_{v_{i-1}}$.
2. Sample $v_i$'s type. If the sampled type is the ending type, stop the decoding, connect all loose ends (nodes without successors) to $v_i$, and output the DAG; otherwise, continue the generation.
3. Update $v_i$'s hidden state by $\mathbf{h}_{v_i} = \mathrm{GRU}_d(\mathbf{x}_{v_i}, \mathbf{h}_{v_i}^{\text{in}})$, where $\mathbf{h}_{v_i}^{\text{in}} = \mathbf{h_0}$ if $i = 1$; otherwise, $\mathbf{h}_{v_i}^{\text{in}}$ is the aggregated message from its predecessors' hidden states given by equation (4).
4. For $j = i-1, i-2, \ldots, 1$: (a) compute the edge probability of $(v_j, v_i)$ using an MLP $f_{\text{add\_edge}}$ based on $\mathbf{h}_{v_j}$ and $\mathbf{h}_{v_i}$; (b) sample the edge; and (c) if a new edge is added, update $\mathbf{h}_{v_i}$ using step 3.

The above steps are iteratively applied to each new generated node, until step 2 samples the ending type. For every new node, we first predict its node type based on the current graph state, and then sequentially predict whether each existing node has a directed edge to it based on the existing and current nodes' hidden states. Figure 3 illustrates this process. Since edges always point to new nodes, the generated graph is guaranteed to be acyclic. Note that we maintain hidden states for both the current node and existing nodes, and keep updating them during the generation. For example, whenever step 4 samples a new edge between $v_j$ and $v_i$, we will update $\mathbf{h}_{v_i}$ to reflect the change of its predecessors and thus the change of the computation so far. Then, we will use the new $\mathbf{h}_{v_i}$ for the next prediction. Such a dynamic updating scheme is flexible, computation-aware, and always uses the up-to-date state of each node to predict next steps. In contrast, methods based on RNNs [3, 13] do not maintain states for old nodes, and only use the current RNN state to predict the next step.

In step 4, when sequentially predicting incoming edges from previous nodes, we choose the reversed order $i - 1, \ldots, 1$ instead of $1, \ldots, i - 1$ or any other order. This is based on the prior knowledge that a new node $v_i$ is more likely to firstly connect from the node $v_{i-1}$ immediately before it. For example, in neural architecture design, when adding a new layer, we often first connect it from the last added layer, and then decide whether there should be skip connections from other previous layers. Note that however, such an order is not fixed and can be flexible according to specific applications.

## 3.3 Training

During the training phase, we use teacher forcing [17] to measure the reconstruction loss: following the topological order with which the input DAG's nodes are consumed, we sum the negative log-likelihood of each decoding step by forcing them to generate the ground truth node type or edge at each step. This ensures that the model makes predictions based on the correct histories. Then, we optimize the VAE loss (the negative of (1)) using mini-batch gradient descent following [17]. Note that teacher forcing is only used in training. During generation, we sample a node type or edge at

each step according to the decoding distributions described in Section 3.2 and calculate subsequent decoding distributions based on the sampled results.

### 3.4 Discussion and model extensions

**Relation with RNNs.** The D-VAE encoder and decoder can be reduced to ordinary RNNs when the input DAG is reduced to a chain of nodes. Although we propose D-VAE from a GNN's perspective, our model can also be seen as a generalization of traditional sequence modeling frameworks [55, 56] where a timestamp depends only on the timestamp immediately before it, to the DAG case where a timestamp has multiple previous dependencies. As special DAGs, similar ideas have been explored for trees [57, 17], where a node can have multiple incoming edges yet only one outgoing edge.

**Bidirectional encoding.** D-VAE's encoding process can be seen as simulating how an input signal goes through a DAG, with $\mathbf{h}_v$ simulating the output signal at each node $v$. This is also known as *forward propagation* in neural networks. Inspired by the bidirectional RNN [58], we can also use another GRU to reversely encode a DAG (i.e., reverse all edge directions and encode the DAG again), thus simulating the *backward propagation* too. After reverse encoding, we get two ending states, which are concatenated and linearly mapped to their original size as the final output state. We find this bidirectional encoding can increase the performance and convergence speed on neural architectures.

**Incorporating vertex semantics.** Note that D-VAE currently uses one-hot encoding of node types as $\mathbf{x}_v$, which does not consider the semantic meanings of different node types. For example, a $3 \times 3$ convolution layer might be functionally very similar to a $5 \times 5$ convolution layer, while being functionally distinct from a max pooling layer. We expect incorporating such semantic meanings of node types to be able to further improve D-VAE's performance. For example, we can use pretrained embeddings of node types to replace the one-hot encoding. We leave it for future work.

## 4 Experiments

We validate the proposed DAG variational autoencoder (D-VAE) on two DAG optimization tasks:

- **Neural architecture search.** Our neural network dataset contains 19,020 neural architectures from the ENAS software [33]. Each neural architecture has 6 layers (excluding input and output layers) sampled from: $3 \times 3$ and $5 \times 5$ convolutions, $3 \times 3$ and $5 \times 5$ depthwise-separable convolutions [59], $3 \times 3$ max pooling, and $3 \times 3$ average pooling. We evaluate each neural architecture's weight-sharing accuracy [33] (a proxy of the true accuracy) on CIFAR-10 [60] as its performance measure. We split the dataset into 90% training and 10% held-out test sets. We use the training set for VAE training, and use the test set only for evaluation.
- **Bayesian network structure learning.** Our Bayesian network dataset contains 200,000 random 8-node Bayesian networks from the `bnlearn` package [61] in R. For each network, we compute the Bayesian Information Criterion (BIC) score to measure the performance of the network structure for fitting the Asia dataset [62]. We split the Bayesian networks into 90% training and 10% test sets. For more details, please refer to Appendix I.

Following [3], we do four experiments for each task:

- **Basic abilities of VAE models.** In this experiment, we perform standard tests to evaluate the reconstructive and generative abilities of a VAE model for DAGs, including reconstruction accuracy, prior validity, uniqueness and novelty.
- **Predictive performance of latent representation.** We test how well we can use the latent embeddings of neural architectures and Bayesian networks to predict their performances.
- **Bayesian optimization.** This is the motivating application of D-VAE. We test how well the learned latent space can be used for searching for high-performance DAGs through Bayesian optimization.
- **Latent space visualization.** We visualize the latent space to qualitatively evaluate its smoothness.

Since there is little previous work on DAG generation, we compare D-VAE with four generative baselines adapted for DAGs: S-VAE, GraphRNN, GCN and DeepGMG. Among them, S-VAE [56] and GraphRNN [13] are adjacency-matrix-based methods; GCN [22] and DeepGMG [6] are graph-based methods which use simultaneous message passing to embed DAGs. We include more details about these baselines and discuss D-VAE's advantages over them in Appendix J. The training details are in Appendix K. All the code and data are available at `https://github.com/muhanzhang/D-VAE`.

Table 1: Reconstruction accuracy, prior validity, uniqueness and novelty (%).

| Methods | Neural architectures | | | | Bayesian networks | | | |
|---|---|---|---|---|---|---|---|---|
| | Accuracy | Validity | Uniqueness | Novelty | Accuracy | Validity | Uniqueness | Novelty |
| D-VAE | 99.96 | 100.00 | 37.26 | 100.00 | 99.94 | 98.84 | 38.98 | 98.01 |
| S-VAE | 99.98 | 100.00 | 37.03 | 99.99 | 99.99 | 100.00 | 35.51 | 99.70 |
| GraphRNN | 99.85 | 99.84 | 29.77 | 100.00 | 96.71 | 100.00 | 27.30 | 98.57 |
| GCN | 98.70 | 99.53 | 34.00 | 100.00 | 99.81 | 99.02 | 32.84 | 99.40 |
| DeepGMG | 94.98 | 98.66 | 46.37 | 99.93 | 47.74 | 98.86 | 57.27 | 98.49 |

## 4.1 Reconstruction accuracy, prior validity, uniqueness and novelty

Being able to accurately reconstruct input examples and generate valid new examples are basic requirements for VAE models. In this experiment, we evaluate the models by measuring 1) how often they can reconstruct input DAGs perfectly (Accuracy), 2) how often they can generate valid neural architectures or Bayesian networks from the prior distribution (Validity), 3) the proportion of unique DAGs out of the valid generations (Uniqueness), and 4) the proportion of valid generations that are never seen in the training set (Novelty).

We first evaluate each model's reconstruction accuracy on the test sets. Following previous work [3, 17], we regard the encoding as a stochastic process. That is, after getting the mean and variance parameters of the posterior approximation $q_\phi(\mathbf{z}|G)$, we sample a $\mathbf{z}$ from it as $G$'s latent vector. To estimate the reconstruction accuracy, we sample $\mathbf{z}$ 10 times for each $G$, and decode each $\mathbf{z}$ 10 times too. Then we report the average proportion of the 100 decoded DAGs that are identical to the input. To calculate prior validity, we sample 1,000 latent vectors $\mathbf{z}$ from the prior distribution $p(\mathbf{z})$ and decode each latent vector 10 times. Then we report the proportion of valid DAGs in these 10,000 generations. A generated DAG is valid if it can be read by the original software which generated the training data. More details about the validity experiment are in Appendix M.1.

We show the results in Table 1. Among all the models, D-VAE and S-VAE generally perform the best. We find that D-VAE, S-VAE and GraphRNN all have near perfect reconstruction accuracy, prior validity and novelty. However, D-VAE and S-VAE show higher uniqueness, meaning that they generate more diverse examples. GCN and DeepGMG have worse reconstruction accuracies for neural architectures due to nonzero training losses. This is because the simultaneous message passing scheme in them focus more on learning local graph structures, but fail to encode the computation represented by the entire neural network. Besides, the sum pooling after the message passing might also lose some global topology information which is important for the reconstruction. The nonzero training loss of DeepGMG acts like an early stopping regularizer, making DeepGMG generate more unique graphs. Nevertheless, reconstruction accuracy is much more important than uniqueness in our tasks, since we want our embeddings to accurately remap to their original structures after latent space optimization.

## 4.2 Predictive performance of latent representation.

In this experiment, we evaluate how well the learned latent embeddings can predict the corresponding DAGs' performances, which tests a VAE's unsupervised representation learning ability. Being able to accurately predict a latent point's performance also makes it much easier to search for high-performance points in this latent space. Thus, the experiment is also an indirect way to evaluate a VAE latent space's amenability for DAG optimization. Following [3], we train a sparse Gaussian process (SGP) model [63] with 500 inducing points on the embeddings of training data to predict the performance of unseen test data. We include the SGP training details in Appendix L.

Table 2: Predictive performance of encoded means.

| Methods | Neural architectures | | Bayesian networks | |
|---|---|---|---|---|
| | RMSE | Pearson's $r$ | RMSE | Pearson's $r$ |
| D-VAE | **0.384±0.002** | **0.920±0.001** | **0.300±0.004** | **0.959±0.001** |
| S-VAE | 0.478±0.002 | 0.873±0.001 | 0.369±0.003 | 0.933±0.001 |
| GraphRNN | 0.726±0.002 | 0.669±0.001 | 0.774±0.007 | 0.641±0.002 |
| GCN | 0.485±0.006 | 0.870±0.001 | 0.557±0.006 | 0.836±0.002 |
| DeepGMG | 0.433±0.002 | 0.897±0.001 | 0.788±0.007 | 0.625±0.002 |

We use two metrics to evaluate the predictive performance of the latent embeddings (given by the mean of the posterior approximations $q_\phi(\mathbf{z}|G)$). One is the RMSE between the SGP predictions and the true performances. The other is the Pearson correlation coefficient (or Pearson's $r$), measuring how well the prediction and real performance tend to go up and down together. A small RMSE and a large Pearson's $r$ indicate a better predictive performance.

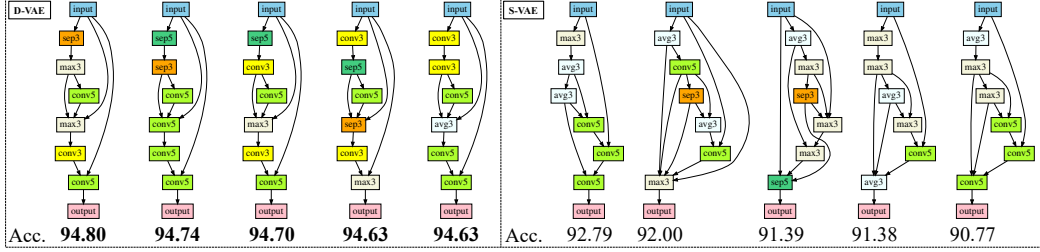

Figure 4: Top 5 neural architectures found by each model and their true test accuracies.

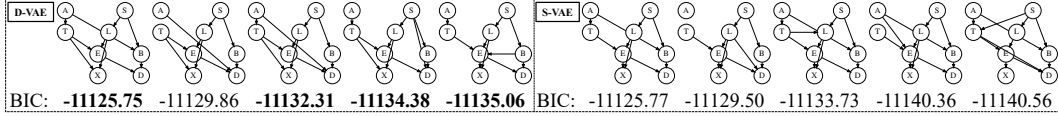

Figure 5: Top 5 Bayesian networks found by each model and their BIC scores (higher the better).

All the experiments are repeated 10 times and the means and standard deviations are reported. Table 2 shows the results. We find that both the RMSE and Pearson's $r$ of D-VAE are significantly better than those of the other models. A possible explanation is that D-VAE encodes the computation, while a DAG's performance is primarily determined by its computation. Therefore, D-VAE's latent embeddings are more informative about performance. In comparison, adjacency-matrix-based methods (S-VAE and GraphRNN) and graph-based methods with simultaneous message passing (GCN and DeepGMG) both only encode (local) graph structures without specifically modeling computations on DAG structures. The better predictive power of D-VAE favors using a predictive model in its latent space to guide the search for high performance graphs.

## 4.3 Bayesian optimization

We perform Bayesian optimization (BO) using the two best models, D-VAE and S-VAE, validated by previous experiments. Based on the SGP model from the last experiment, we perform 10 iterations of batch BO, and average results across 10 trials. Following Kusner et al. [3], in each iteration, a batch of 50 points are proposed by sequentially maximizing the expected improvement (EI) acquisition function, using Kriging Believer [64] to assume labels for previously chosen points in the batch. For each batch of selected points, we evaluate their decoded DAGs' real performances and add them back to the SGP to select the next batch. Finally, we check the best-performing DAGs found by each model to evaluate its DAG optimization performance.

**Neural architectures.** For neural architectures, we select the top 15 found architectures in terms of their weight-sharing accuracies, and fully train them on CIFAR-10's train set to evaluate their true test accuracies. More details can be found in Appendix H. We show the 5 architectures with the highest true test accuracies in Figure 4. As we can see, D-VAE in general found much better neural architectures than S-VAE. Among the selected architectures, D-VAE achieved a highest accuracy of 94.80%, while S-VAE's highest accuracy was only 92.79%. In addition, all the 5 architectures of D-VAE have accuracies higher than 94%, indicating that D-VAE's latent space can stably find many high-performance architectures. More details about our NAS experiments are in Appendix H.

**Bayesian networks.** We similarly report the top 5 Bayesian networks found by each model ranked by their BIC scores in Figure 5. D-VAE generally found better Bayesian networks than S-VAE. The best Bayesian network found by D-VAE achieved a BIC of -11125.75, which is better than the best network in the training set with a BIC of -11141.89 (a higher BIC score is better). Note that BIC is in log scale, thus the probability of our found network to explain the data is actually 1E7 times larger than that of the best training network. For reference, the true Bayesian network used to generate the Asia data has a BIC of -11109.74. Although we did not exactly find the true network, our found network was close to it and outperformed all 180,000 training networks. Our experiments show that searching in an embedding space is a promising direction for Bayesian network structure learning.

## 4.4 Latent space visualization

In this experiment, we visualize the latent spaces of the VAE models to get a sense of their smoothness.

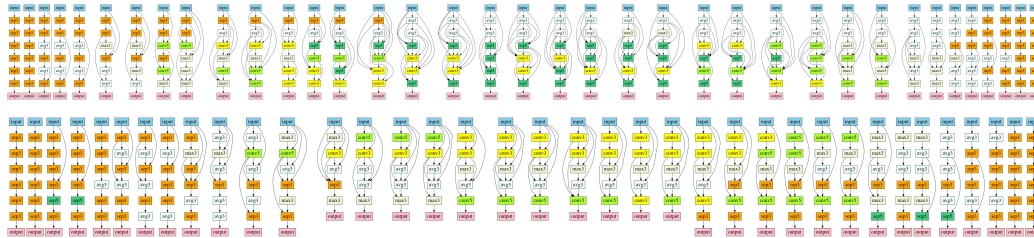

Figure 6: Great circle interpolation starting from a point and returning to itself. Upper: D-VAE. Lower: S-VAE.

For neural architectures, we visualize the decoded architectures from points along a great circle in the latent space [65] (slerp). We start from the latent embedding of a straight network without skip connections. Imagine this latent embedding as a point on the surface of a sphere (visualize the earth). We randomly pick a great circle starting from this point and returning to itself around the sphere. Along this circle, we evenly pick 35 points and visualize their decoded neural architectures in Figure 6. As we can see, both D-VAE and S-VAE show relatively smooth interpolations by changing only a few node types or edges each time. Visually speaking, S-VAE's structural changes are even smoother. This is because S-VAE treats DAGs as strings, thus tending to embed DAGs with few differences in string representations to similar regions of the latent space without considering their computational differences (see Appendix J for more discussion of this problem). In contrast, D-VAE models computations, and focuses more on the smoothness w.r.t. computation rather than structure.

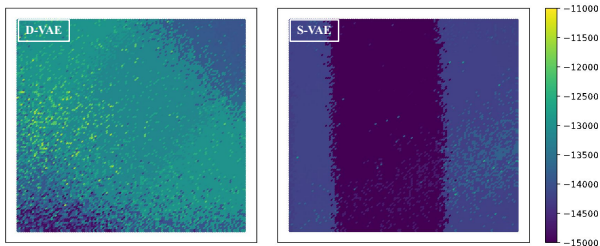

Figure 7: Visualizing a principal 2-D subspace of the latent space.

For Bayesian networks, we directly visualize the BIC score distribution of the latent space. To do so, we reduce its dimensionality by choosing a 2-D subspace spanned by the first two principal components of the training data's embeddings. In this low-dimensional subspace, we compute the BIC scores of all the points evenly spaced within a $[-0.3, 0.3]$ grid and visualize the scores using a colormap in Figure 7. As we can see, D-VAE seems to better differentiate high-score points from low-score ones and shows more smoothly changing BIC scores. In comparison, S-VAE shows sharp boundaries and seems to mix high-score and low-score points more severely. The smoother latent space might be the key reason for the better Bayesian optimization performance with D-VAE. Furthermore, we notice that D-VAE's 2-D latent space is brighter; one explanation is the two principal components of D-VAE explain more variance (59%) of training data than those of S-VAE (17%). Thus, along the two principal components of S-VAE we will see less points from the training distribution. These out-of-distribution points tend to decode to not very good Bayesian networks, thus are darker. This also indicates that D-VAE learns a more compact latent space.

## 5 Conclusion

In this paper, we have proposed D-VAE, a GNN-based deep generative model for DAGs. D-VAE uses a novel asynchronous message passing scheme to encode a DAG respecting its partial order, which explicitly models the computations on DAGs. By performing Bayesian optimization in D-VAE's latent spaces, we offer promising new directions to two important problems, neural architecture search and Bayesian network structure learning. We hope D-VAE can inspire more research on studying DAGs and their applications in the real world.

### Acknowledgments

MZ, ZC and YC were supported by the National Science Foundation (NSF) under award numbers III-1526012 and SCH-1622678, and by the National Institute of Health under award number 1R21HS024581. SJ and RG were supported by the NSF under award numbers IIA–1355406, IIS–1845434, and OAC–1940224. The authors would like to thank Liran Wang for the helpful discussions.

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
