[Supplementary Material]

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

# Appendices

## A   More Related Work

Both neural architecture search (NAS) and Bayesian network structure learning (BNSL) are subfields of AutoML. See Zöller and Huber [66] for a survey. We have given a brief overview of NAS and BNSL in Section 2. Below we discuss several works most related to our work in more detail.

Luo et al. [39] proposed a novel NAS approach called Neural Architecture Optimization (NAO). The basic idea is to jointly learn an encoder-decoder between networks and a *continuous* space, and also a performance predictor $f$ that maps the continuous representation of a network to its performance on a given dataset; then they perform two or three iterations of gradient descent on $f$ to find better architectures in the continuous space, which are then decoded to real networks to evaluate. This methodology is similar to that of Gómez-Bombarelli et al. [2] and Jin et al. [17] for molecule optimization; also similar to Mueller et al. [67] for slightly revising a sentence.

There are several key differences comparing to our approach. First, NAO uses strings (e.g. "node-2 conv 3x3 node1 max-pooling 3x3") to represent neural architectures, whereas we directly use graph representations, which is more natural and generally applicable to other graphs such as Bayesian network structures. Second, NAO uses supervised learning instead of unsupervised learning, which means it needs to first evaluate a considerable amount of randomly sampled graphs on a typically large dataset (e.g. train many neural networks), and use these results to supervise the training of the autoencoder. Given a new dataset, the autoencoder needs to be completely retrained. In contrast, we train our variational autoencoder in a fully unsupervised manner, so the model is of general purposes.

Fusi et al. [68] proposed a novel AutoML algorithm also using model embedding, but with a matrix factorization approach. They first construct a matrix of performances of thousands of ML pipelines on hundreds of datasets; then they use a probabilistic matrix factorization to get the latent representations of the pipelines. Given a new dataset, Bayesian optimization with the expected improvement heuristic is used to find the best pipeline. This approach only allows us to choose from predefined off-the-shelf ML models, hence its flexibility is somewhat limited.

Kandasamy et al. [35] use Bayesian optimization for NAS; they define a kernel that measures the similarities between networks by solving an optimal transport problem, and in each iteration, they use some evolutionary heuristics to generate a set of candidate networks based on making small modifications to existing networks, and use expected improvement to choose the next one to evaluate. This work is similar to ours in terms of the application of Bayesian optimization. However, defining a kernel to measure the similarities between discrete structures is a non-trivial problem. In addition, the discrete search space is heuristically extrapolated near existing architectures, which makes the search essentially local. In contrast, we directly fit a Gaussian process over the entire continuous latent space, enabling more global optimization.

Using Gaussian process (GP) for Bayesian network structure learning has also been studied before. Yackley and Lane [69] analyzed the smoothness of BDe score, showing that a local change (e.g. adding an edge) can change the score by at most $\mathcal{O}(\log n)$, where $n$ is the number of training points. They proposed to use GP as a proxy for the score to accelerate the search. Anderson and Lane [70] used GP to model the BDe score, and showed that the probability of improvement is better than that of using hill climbing to guide the local search. However, these methods still heuristically and locally operate in the discrete space, whereas our latent space makes both local and global methods such as gradient descent and Bayesian optimization applicable in a principled manner.

Recently, Zheng et al. [50] also proposed a continuous optimization approach for BNSL, where the decision variable is the adjacency matrix of the DAG and the objective function is least square loss based on linear structural equation modeling (SEM); acyclicity is ensured by a novel equality constraint. Yu et al. [51] generalize this approach to nonlinear SEM using VAE. We highlight several key differences from our approach: 1) they directly optimize the adjacency matrix, but we optimize a learned latent representation of the DAGs; 2) they ensure acyclicity by enforcing an equality constraint, but for our method acyclicity is automatically guaranteed by the decoding process; 3) They use gradient-based optimization, but we use global black-box optimization. 4) Their methods are specific to BNSL, but ours applies to general DAG optimization; 5) The usage of VAE is totally different: they

use VAE as a generative model for the data (sampled from the DAG), but we use VAE as a generative model for DAGs.

## B  Graph Structure vs. Computation vs. Function

In Section 3 we defined computation. Here we discuss the differences among DAG structure, computation and function. A DAG structure with operations on nodes define a computation, and two DAGs can define the same computation, which are illustrated in Figure 1. A computation defines a function, and two computations can define the same function. For example, computation $C_1 := x + 1 - 1$ defines a function $f(x) = x$, while computations $C_2 := x - 1 + 1$ and $C_3 := x$ also define the function $f(x) = x$. However, $C_1$, $C_2$ and $C_3$ are different computations. In other words, a computation is (informally speaking) a process which focuses on the course of how the input is processed into the output, while a function is a mapping from input to output which does not care about the process.

Sometimes, the same computation can also define different functions, e.g., two identical neural architectures will represent different functions given they are trained differently (since the weights of their layers will be different). In D-VAE, we model computations instead of functions, since 1) modeling functions is much harder than modeling computations (requires understanding the semantic meaning of each operation, such as the cancelling out of $+$ and $-$), and 2) modeling functions additionally requires knowing the parameters of some operations, which are unknown before training.

Note also that in Definition 1, we only allow one single input signal. But in real world a computation sometimes has multiple initial input signals. However, the case of multiple input signals can be reduced to the single input case by adding an initial assignment operation that assigns the combined input signal to their corresponding next-level operations. For ease of presentation, we uniformly assume single input throughout the paper.

## C  Proof of Theorem 1

*Proof.* Let $v_1$ be the starting node with no predecessors. By assumption, $v_1$ is the single starting node no matter how we permute the nodes of the input DAG. For $v_1$, the aggregation function $\mathcal{A}$ always outputs a zero vector. Thus, $\mathbf{h}_{v_1}^{\text{in}}$ is invariant to node permutations. Subsequently, the hidden state $\mathbf{h}_{v_1} = \mathcal{U}(\mathbf{x}_{v_1}, \mathbf{h}_{v_1}^{\text{in}})$ is also invariant to node permutations.

Now we prove the theorem by structural induction. Consider node $v$. Suppose for every predecessor $u$ of $v$, the hidden state $\mathbf{h}_u$ is invariant to node permutations. We will show that $\mathbf{h}_v$ is also invariant to node permutations. Notice that in (3), the output $\mathbf{h}_v^{\text{in}}$ by $\mathcal{A}$ is invariant to node permutations, since $\mathcal{A}$ is invariant to the order of its inputs $\mathbf{h}_u$, and all $\mathbf{h}_u$ are invariant to node permutations. Subsequently, node $v$'s hidden state $\mathbf{h}_v = \mathcal{U}(\mathbf{x}_v, \mathbf{h}_v^{\text{in}})$ is invariant to node permutations. By induction, we know that every node's hidden state is invariant to node permutations, including the ending node's hidden state. Thus, the D-VAE encoder is invariant to node permutations. $\square$

## D  Proof of Theorem 2

*Proof.* Suppose there is an arbitrary input signal $x$ fed to the starting node $v_1$. For convenience, we will use $C_i(x)$ to denote the output signal at vertex $v_i$, where $C_i$ represents the composition of all the operations along the paths from $v_1$ to $v_i$.

For the starting node $v_1$, remember we feed a fixed $\mathbf{h}_{v_1}^{\text{in}} = \mathbf{0}$ to (2), thus $\mathbf{h}_{v_1}$ is also fixed. Since $C_1$ also represents a fixed input operation, we know that the mapping from $C_1$ to $\mathbf{h}_{v_1}$ is injective. Now we prove the theorem by induction. Assume the mapping from $C_j$ to $\mathbf{h}_{v_j}$ is injective for all $1 \leq j < i$. We will prove that the mapping from $C_i$ to $\mathbf{h}_{v_i}$ is also injective.

Let $\phi_j(C_j) = \mathbf{h}_{v_j}$ where $\phi_j$ is injective. Consider the output signal $C_i(x)$, which is given by feeding $\{C_j(x) : v_j \to v_i\}$ to $o_i$. Thus,

$$C_i(x) = o_i(\{C_j(x) : v_j \to v_i\}). \tag{6}$$

In other words, we can write $C_i$ as

$$C_i = \psi(o_i, \{C_j : v_j \to v_i\}), \tag{7}$$

where $\psi$ is an injective function used for defining the composite computation $C_i$ based upon $o_i$ and $\{C_j : v_j \to v_i\}$. Note that $\{C_j : v_j \to v_i\}$ can be either unordered or ordered depending on the operation $o_i$. For example, if $o_i$ is some symmetric operations such as adding or multiplication, then $\{C_j : v_j \to v_i\}$ can be unordered. If $o_i$ is some operation like subtraction or division, then $\{C_j : v_j \to v_i\}$ must be ordered.

With (2) and (3), we can write the hidden state $\mathbf{h}_{v_i}$ as follows:

$$\mathbf{h}_{v_i} = \mathcal{U}(\mathbf{x}_{v_i}, \mathcal{A}(\{\mathbf{h}_{v_j} : v_j \to v_i\}))$$
$$= \mathcal{U}(O(o_i), \mathcal{A}(\{\phi_j(C_j) : v_j \to v_i\})), \tag{8}$$

where $O$ is the injective one-hot encoding function mapping $o_i$ to $\mathbf{x}_{v_i}$. In the above equation, $\mathcal{U}, O, \mathcal{A}, \phi_j$ are all injective. Since the composition of injective functions is injective, there exists an injective function $\varphi$ so that

$$\mathbf{h}_{v_i} = \varphi(o_i, \{C_j : v_j \to v_i\}). \tag{9}$$

Then combining (7) we have:

$$\mathbf{h}_{v_i} = \varphi \circ \psi^{-1}\psi(o_i, \{C_j : v_j \to v_i\})$$
$$= \varphi \circ \psi^{-1}(C_i). \tag{10}$$

$\varphi \circ \psi^{-1}$ is injective since the composition of injective functions is injective. Thus, we have proved that the mapping from $C_i$ to $\mathbf{h}_{v_i}$ is injective. $\qquad\square$

## E  Modifications for Encoding Neural Architectures

According to Theorem 2, to ensure D-VAE injectively encodes computations, we need the aggregation function $\mathcal{A}$ to be injective. Remember $\mathcal{A}$ takes the multiset $\{\mathbf{h}_u : u \to v\})$ as input. If the order of its elements does not matter, then the gated sum in (4) can model this injective multiset function without issues. However, if the order matters (i.e., permuting the elements of $\{\mathbf{h}_u : u \to v\}$ makes $\mathcal{A}$ output different results), we need a different aggregation function that can encode such orders.

Whether the order should matter for $\mathcal{A}$ depends on whether the input order matters for the operations $o$ (see the proof for Theorem 2 for more details). For example, if multiple previous layers' outputs are summed or averaged as the input to a next layer in the neural networks, then $\mathcal{A}$ can be modeled by the gated sum in (4) as the order of inputs does not matter. However, if these outputs are concatenated as the next layer's input, then the order does matter. In our experiments, the neural architectures use the second way to aggregate outputs from previous layers. The order of concatenation depends on a global order of the layers in a neural architecture. For example, if layer-2 and layer-4's outputs are input to layer-5, then layer-2's output will be before layer-4's output in their concatenation.

Since the gated sum in (4) can only handle the unordered case, we can slightly modify (4) in order to make it order-aware thus more suitable for our neural architectures. Our scheme is as follows:

$$\mathbf{h}_v^{\text{in}} = \sum_{u \to v} g(\text{Concat}(\mathbf{h}_u, \mathbf{x}_{\text{uid}})) \odot m(\text{Concat}(\mathbf{h}_u, \mathbf{x}_{\text{uid}})), \tag{11}$$

where $\mathbf{x}_{\text{uid}}$ is the one-hot encoding of layer $u$'s global ID (1,2,3,...). Such an aggregation function respects the concatenation order of the layers. We empirically observed that this aggregation function can increase D-VAE's performance on neural architectures compared to the plain aggregation function (4). However, even using (4) still outperformed all baselines.

## F  Modifications for Encoding Bayesian Networks

We also make some modifications when encoding Bayesian networks. One modification is that the aggregation function (4) is changed to:

$$\mathbf{h}_v^{\text{in}} = \sum_{u \to v} g(\mathbf{x}_u) \odot m(\mathbf{x}_u). \tag{12}$$

Compared to (4), we replace $\mathbf{h}_u$ with the node type feature $\mathbf{x}_u$. This is due to the differences between computations on a neural architecture and on a Bayesian network. In a neural network, the signal flow follows the network architecture, where the output signal of a layer is fed as the input signals to its succeeding layers. Also in a neural network, what we are interested in is the result output by the final layer. In contrast, for a Bayesian network, the graph represents a set of conditional

dependencies among variables instead of a computational flow. In particular, for Bayesian network structure learning, we are often concerned about computing the (log) marginal likelihood score of a dataset given a graph structure, which is often decomposed into individual variables given their parents (see Definition 18.2 in Koller and Friedman [1]). For example, in Figure 8, the overall score can be decomposed into $s(X_1) + s(X_2) + s(X_3 \mid X_1, X_2) + s(X_4) + s(X_5 \mid X_3, X_4)$. To compute the score $s(X_5 \mid X_3, X_4)$ for $X_5$, we only need the values of $X_3$ and $X_4$; its grandparents $X_1$ and $X_2$ should have no influence on $X_5$. Based on this intuition, when computing the hidden state of a

Figure 8: An example Bayesian network and its encoding.

node, we use the features $\mathbf{x}_u$ of its parents $u$ instead of $\mathbf{h}_u$, which "d-separates" the node from its grandparents. For the update function, we still use (5).

Also based on the decomposibility of the score, we make another modification for encoding Bayesian networks by using the sum of all node states as the final output state instead of only using the ending node state. Similarly, when decoding Bayesian networks, the graph state $\mathbf{h}_G := \sum_{j=1,\dots,i-1} \mathbf{h}_{v_j}$.

Note that the combination of (12) and (5) can injectively model the conditional dependence between $v$ and its parents $u$. In addition, using summing can model injective set functions [53, Lemma 5]. Therefore, the above encoding scheme is able to **injectively encode** the complete **conditional dependencies** of a Bayesian network, thus also the overall score function $s$ of the network.

# G    Advantages of Encoding Computations in DAG Optimization

Here we discuss why D-VAE's ability to injectively encode computations (Theorem 2) is of great benefit to performing DAG optimization in the latent space. Firstly, our target is to find a DAG that achieves high performance (e.g., accuracy of neural network, BIC score of Bayesian network) on a given dataset. The performance of a DAG is directly related to its computation. For example, given the same set of layer parameters, two neural networks with the same computation will have the same performance on a given test set. Since D-VAE encodes computations instead of structures, it allows **embedding DAGs with similar performances to the same regions** in the latent space, rather than embedding DAGs with merely similar structure patterns to the same regions. Subsequently, the latent space can be **smooth w.r.t.** *performance* instead of *structure.* Such smoothness can greatly facilitate searching for high-performance DAGs in the latent space, since similar-performance DAGs tend to locate near each other in the latent space instead of locating randomly, and modeling a smoothly-changing performance surface is much easier.

Note that Theorem 2 is a necessary condition for the latent space to be smooth w.r.t. performance, because if D-VAE cannot injectively encode computations, it might map two DAGs representing completely different computations to the same encoding, making this point of the latent space arbitrarily unsmooth. Although there yet is no theoretical guarantee that the latent space must be smooth w.r.t. DAGs' performances, we do empirically observe that the predictive performance and Bayesian optimization performance of D-VAE's latent space are significantly better than those of baselines, which is indirect evidence that D-VAE's latent space is smoother w.r.t. performance. Our visualization results also confirm the smoothness. See Section 4.2, 4.3, 4.4 for details.

# H    More Details about Neural Architecture Search

We use the efficient neural architecture search (ENAS)'s software [33] to generate the training and testing neural architectures. With these seed architectures, we can train a VAE model and thus search for new high-performance architectures in the latent space.

ENAS alternately trains two components: 1) an RNN-based controller which is used to propose new architectures, and 2) the shared weights of the proposed architectures. It uses a weight-sharing

scheme to obtain a quick but rough estimate of how good an architecture is. That is, it forces all the proposed architectures to use the same set of shared weights, instead of fully training each neural network individually. It assumes that an architecture with a high validation accuracy using the shared weights (i.e., the weight-sharing accuracy) is more likely to have a high test accuracy after fully retraining its weights from scratch.

We first run ENAS in the macro space (Section 2.3 of [33]) for 1000 epochs with 20 architectures proposed in each epoch. For all the proposed architectures excluding the first 1000 burn-in ones, we evaluate their weight-sharing accuracies using the shared weights from the last epoch. We further split the data into 90% training and 10% held-out test sets. Then our task becomes to train a VAE on the training neural architectures, and then generate new high-performance architectures from the latent space based on Bayesian optimization. Note that our target performance measure here is the weight-sharing accuracy, not the true validation/test accuracy after fully retraining the architecture. This is because the weight-sharing accuracy takes around 0.5 second to evaluate, while fully training a network takes over 12 hours. In consideration of our limited computational resources, we choose the weight-sharing accuracy as our optimization target in the Bayesian optimization experiments.

After the Bayesian optimization finds a final set of architectures with high weight-sharing accuracies, we will fully train them to evaluate their true test accuracies on CIFAR-10. To fully train an architecture, we follow the original setting of [33] to train each architecture on CIFAR-10's training set for 310 epochs, and report the last epoch's net's test accuracy. See [33, Section 3.2] for details.

Due to our constrained computational resources, we choose not to perform Bayesian optimization to optimize the true validation accuracy (obtained by fully training a neural network), which would be a more principled way for searching neural architectures. Nevertheless, we describe its procedure here for future explorations: After training the D-VAE, we have no architectures at all to initialize a Gaussian process regression on the true validation accuracy. Thus, we need to randomly pick up some points in the latent space, decode them into neural architectures, and get their true validation accuracies after full training. Then with these initial points, we start the Bayesian optimization similarly to Section 4.3, with the optimization target replaced by the true validation accuracy. Finally, we will find a set of architectures with the highest true validation accuracies, and report their true test accuracies. This experiment will take much longer time (possibly months of GPU time). Thus, it is very necessary to train multiple models parallelly on many machines, like [28] does.

One might wonder why we train another generative model after we already have ENAS. Firstly, ENAS is a task-specific supervised model. It leverages the validation accuracy signals of the target task to guide the generation of new architectures based on reinforcement learning. For any new NAS task, ENAS needs to be completely retrained. In contrast, D-VAE is unsupervised. Once trained, it can be applied to NAS tasks targeting different datasets. For example, although we use the neural architectures generated by the ENAS targeting CIFAR-10 to train our D-VAE, once trained, we can use D-VAE's latent space to search neural architectures suitable for CV tasks other than CIFAR-10. In contrast, the trained ENAS is not applicable to other tasks since it uses supervised signals from CIFAR-10. In other words, ENAS is only used to generate a set of seed architectures for training D-VAE, and is not necessary for D-VAE. For example, we may also train D-VAE using the recent NAS-Bench-101 dataset[1], which we leave for future work. Another exclusive advantage of D-VAE is that it provides a way to learn neural architecture embeddings, which can be used for downstream tasks such as visualization and classification, etc.

In the Bayesian optimization experiments (Section 4.3), the best architecture found by D-VAE achieves a test accuracy of 94.80% on CIFAR-10. Although not outperforming state-of-the-art NAS techniques such as NAONET which has an error rate of 2.11%, our architecture only contains 3 million parameters compared to NAONET + CUTOUT which has 128 million parameters [39]. In addition, the search space is different between the two approaches: we directly search 6-layer CNNs, while NAONET searches 5-layer CNN cells and stacks the found cell for 6 times to construct a CNN, thus having much deeper final networks. Finally, NAONET used 200 GPUs to fully train 1,000 architectures for 1 day, and added 4 times more filters after optimization. In comparison, we only used 1 GPU to evaluate the weight-sharing accuracy, and did not add filters to boost the performance. We emphasize that the main purpose of the paper is to introduce a DAG generative model that is capable of DAG optimization, rather than to break NAS records.

# I   More Details about Bayesian Network Structure Learning

We consider a small synthetic problem called Asia [62] as our target Bayesian network structure learning problem. The Asia dataset is composed of 5,000 samples, each is generated by a true network with 8 binary variables[2]. Bayesian Information Criteria (BIC) score is used to evaluate how well a Bayesian network fits the 5,000 samples. To train a VAE model to generate Bayesian network structures, we sample 200,000 random 8-node Bayesian networks from the `bnlearn` package [61] in R, which are split into 90% training and 10% testing sets. Our task is to train a VAE model on the training Bayesian networks, and search in the latent space for Bayesian networks with high BIC scores using Bayesian optimization. In this task, we consider a simplified case where the topological order of the true network is known – we let the sampled training and test Bayesian networks have topological orders consistent with the true network of Asia. This is a reasonable assumption for many practical applications, e.g., when the variables have a temporal order [1]. When sampling a network, the probability of a node having an edge with a previous node (as specified by the order) is set to the default option $2/(k-1)$, where $k = 8$ is the number of nodes, which results in sparse graphs where the number of edges is in the same order of the number of nodes.

# J   Baselines

As discussed in the related work, there are other types of graph generative models that can potentially work for DAGs. We explore three possible approaches and contrast them with D-VAE.

**S-VAE.** The S-VAE baseline treats a DAG as a sequence of node strings, which we call string-based variational autoencoder (S-VAE). In S-VAE, each node is represented as the one-hot encoding of its type number concatenated with a 0/1 indicator vector indicating which previous nodes have directed edges to it (i.e., a column of the adjacency matrix). For example, suppose there are two node types and five nodes, then node 4's string "0 1, 0 1 1 0 0" means this node has type 2, and has directed edges from previous nodes 2 and 3. S-VAE leverages a standard GRU-based RNN variational autoencoder [56] on the topologically sorted node sequences, with each node's string treated as its input bit vector.

**GraphRNN.** One similar generative model is GraphRNN [13]. Different from S-VAE, it further decomposes an adjacency column into entries and generates the entries one by one using another edge-level GRU. GraphRNN is a pure generative model which does not have an encoder, thus cannot optimize DAG performance in a latent space. To compare with GraphRNN, we equip it with S-VAE's encoder and use it as another baseline. Note that the original GraphRNN feeds nodes using a BFS order (for undirected graphs), yet we find that it is much worse than using a topological order here. Note also that although GraphRNN seems more expressive than S-VAE, we find that in our applications GraphRNN tends to have more severe overfitting and generates less diverse DAGs.

Figure 9: Two bits of change in the string representations can completely change the computational purpose.

Both GraphRNN and S-VAE treat DAGs as bit strings and use RNNs to model them. This representation has several drawbacks. Firstly, since the topological ordering is often not unique for a DAG, there might be multiple string representations for the same DAG, which all result in different encoded representations. This will violate the permutation invariance in Theorem 1. Secondly, the string representations can be very brittle in terms of modeling DAGs' computational purposes. In Figure 9, the left and right DAGs' string representations are only different by two bits, i.e., the edge (2,3) in the left is changed to the edge (1,3) in the right. However, the two bits of change in structure greatly changes the signal flow, which makes the right DAG always output 1. In S-VAE and GraphRNN, since the bit representations of the left and right DAGs are very similar, they are highly likely to be encoded to similar latent vectors. In particular, the only difference between encoding the left and right DAGs is that, for node 3, the encoder RNN will read an adjacency column of [0, 1, 0, 0, 0, 0] in the left, and read [1, 0, 0, 0, 0, 0] in the right, while all the remaining encoding is exactly the same. By embedding two DAGs serving very different computational purposes to the same region of the latent space, S-VAE and GraphRNN tend to have less smooth latent spaces which make optimization

on them more difficult. In contrast, D-VAE can better differentiate such subtle differences, as the change of edge (2,3) to (1,3) completely changes what aggregated message node 3 receives in D-VAE (hidden state of node 2 vs. hidden state of node 1), which greatly affects node 3 and all its successors' feature learning.

**GCN.** The graph convolutional network (GCN) [22] is one representative graph neural network with a simultaneous message passing scheme. In GCN, all the nodes take their neighbors' incoming messages to update their own states simultaneously instead of following an order. After message passing, the summed node states is used as the graph state. We include GCN as the third baseline. Since GCN can only encode graphs, we equip GCN with D-VAE's decoder to make it a VAE model. For neural architectures, we searched the number of message passing layers from 1 to 5. We found that if we only use 1 message passing layer, the reconstruction accuracy is only around 5%. And if we use 2 or more layers, the reconstruction accuracy gets around 97% stably but never reaches nearly 100% like other models. This demonstrates GCN's limitation of only encoding local substructures for neural architectures. The final GCN model uses 3 message passing layers. For Bayesian networks, we find 1 layer is enough to reach a 99% reconstruction accuracy, which is reasonable since Bayesian networks are naturally local. We report a GCN model using 2 message passing layers.

Using GCN as the encoder can ensure permutation invariance, since node ordering does not matter in GCN. However, GCN's message passing focuses on propagating the neighboring nodes' features to each center node to encode the **local substructure pattern** around each node. In comparison, D-VAE's message passing simulates how the computation is performed along the directed paths of a DAG and focuses on encoding the computation. Although learning local structural features is essential for GCN's successes in node classification and graph classification, here in our tasks, modeling the computation represented by the entire graph is much more important than modeling the local features. Encoding only local substructures may also lose important information about the global DAG topology, making it more difficult to reconstruct the DAG.

**DeepGMG.** DeepGMG [6] is a graph-based graph generative model that uses a simultaneous message passing to learn intermediate node/graph states and uses a similar decoding scheme to D-VAE to generate nodes/edges of a graph sequentially. DeepGMG is originally designed for generating general (undirected) graphs. Several modifications are made to adapt it to our tasks. First, we make it a VAE by equipping it with a 3-layer message passing network as the encoder using its own message passing functions, and use the original generative model as the decoder. Second, we feed in nodes using a topo-order instead of the original random order (and see much improvement). Third, we make the sampled edges in the decoding phase only point to new nodes to ensure acyclicity.

Similar to GCN, DeepGMG's training loss never reaches near zero even with extensive hyperparameter tuning, which again reveals the limitation of simultatenous message passing for encoding DAGs. In comparison, D-VAE can be perfectly trained to near zero loss.

We omit other possible approaches such as GraphVAE [12] and some recent graph-based models [17, 18, 19] etc., either because they lack official code or they target specific graphs (such as molecules) only.

# K VAE Training Details

We use the same settings and hyperparameters (where applicable) for all the four models to be as pair as possible. Many hyperparameters are inherited from Kusner et al. [3]. Single-layer GRUs are used in all models requiring recurrent units, with the same hidden state size of 501. We set the dimension of the latent space to be 56 for all models. All VAE models use $\mathcal{N}(\mathbf{0}, \mathbf{I})$ as the prior distribution $p(\mathbf{z})$, and take $q_\phi(\mathbf{z}|G)$ ($G$ denotes the input DAG) to be a normal distribution with a diagonal covariance matrix, whose mean and variance parameters are output by the encoder. The two MLPs used to output the mean and variance parameters are all implemented as single linear layer networks.

For the decoder network of D-VAE, we let $f_{\text{add\_vertex}}$ and $f_{\text{add\_edge}}$ be two-layer MLPs with ReLU nonlinearities, where the hidden layer sizes are set to two times of the input sizes. Softmax activation is used after $f_{\text{add\_vertex}}$, and sigmoid activation is used after $f_{\text{add\_edge}}$. For the gating network $g$, we use a single linear layer with sigmoid activation. For the mapping function $m$, we use a linear mapping without activation. The bidirectional encoding discussed in Section 3.4 is enabled for D-VAE on

neural architectures, and disabled for D-VAE on Bayesian networks and other models where it gets no better results.

When optimizing the VAE loss, we use ReconstructLoss $+ \alpha$KLDivergence as the loss function. In the original VAE framework, $\alpha$ is set to 1. However, we found that it led to poor reconstruction accuracies, similar to the findings of previous work [3, 10, 17]. Following the implementation of Jin et al. [17], we set $\alpha = 0.005$. Mini-batch SGD with Adam optimizer [71] is used for all models. For neural architectures, we use a batch size of 32 and train all models except DeepGMG for 300 epochs. For Bayesian networks, we use a batch size of 128 and train all models except DeepGMG for 100 epochs. For DeepGMG, we early stop the training at epoch 30 and epoch 5 for neural architectures and Bayesian networks, respectively, in order to avoid numerical instabilities. We use an initial learning rate of 1E-4, and multiply the learning rate by 0.1 whenever the training loss does not decrease for 10 epochs. We use PyTorch to implement all the models.

## L   SGP Training Details

We use sparse Gaussian process (SGP) regression as the predictive model. We use the open sourced SGP implementation in [3]. Both the training and testing data's performances are standardized according to the mean and std of the training data's performances before feeding to the SGP. And the RMSE and Pearson's $r$ in Table 2 are also calculated on the standardized performances. We use the default Adam optimizer to train the SGP for 100 epochs constantly with a mini-batch size of 1,000 and learning rate of 5E-4.

For neural architectures, we use all the training data to train the SGP. For Bayesian networks, we randomly sample 5,000 training examples each time, due to two reasons: 1) using all the 180,000 examples to train the SGP might not be realistic for a typical scenario where network/dataset is large and evaluating a network is expensive; and 2) we found using a smaller sample of training data results in more stable BO performance due to the less probability of duplicate rows which might result in ill conditioned matrices. Note also that, when training the variational autoencoders, all the training data are used, since the VAE training is purely unsupervised.

## M   More Experimental Results

### M.1   More details on the piror validity experiment

Since different models can have different levels of convergence w.r.t. the KLD loss in (1), their posterior distribution $q_\phi(\mathbf{z} \mid \mathbf{x})$ may have different degrees of alignment with the prior distribution $p(\mathbf{z}) = \mathcal{N}(\mathbf{0}, \mathbf{I})$. If we evaluate prior validity by sampling from $p(\mathbf{z})$ for all models, we will favor those models that have a higher-level of KLD convergence. To remove such effects and focus purely on models' intrinsic ability to generate valid DAGs, when evaluating prior validity, we apply $\mathbf{z} = \mathbf{z} \odot \text{std}(\mathbf{Z}_{\text{train}}) + \text{mean}(\mathbf{Z}_{\text{train}})$ for each model (where $\mathbf{Z}_{\text{train}}$ are encoded means of the training data by the model), so that the latent vectors are scaled and shifted to the center of the training data's embeddings. If we do not apply such transformations, we find that we can easily control the prior validity results by optimizing for more or less epochs or putting more or less weight on the KLD loss.

For a generated neural architecture to be read by ENAS, it has to pass the following validity checks: 1) It has one and only one starting node (the input layer); 2) It has one and only one ending type (the output layer); 3) Other than the input node, there are no nodes which do not have any predecessors (no isolated paths); 4) Other than the output node, there are no nodes which do not have any successors (no blocked paths); 5) Each node must have a directed edge from the node immediately before it (the constraint of ENAS), i.e., there is always a main path connecting all the nodes; and 6) It is a DAG.

For a generated Bayesian network to be read by `bnlearn` and evaluated on the Asia dataset, it has to pass the following validity checks: 1) It has exactly 8 nodes; 2) Each type in "ASTLBEXD" appears exactly once; and 3) It is a DAG.

Note that the training graphs generated by the original software all satisfy these validity constraints.

Figure 10: Comparing BO with random search on neural architectures. Left: average weight-sharing accuracy of the selected points in each iteration. Right: highest weight-sharing accuracy of the selected points over time.

Figure 11: Comparing BO with random search on Bayesian networks. Left: average BIC score of the selected points in each iteration. Right: highest BIC score of the selected points over time.

## M.2 Bayesian optimization vs. random search

To validate that Bayesian optimization (BO) in the latent space does provide guidance in searching better DAGs, we compare BO with Random (which randomly samples points from the latent space of D-VAE). Figure 10 and 11 show the results (averaged across 10 trials). In each figure, the left plot shows the average performance of all the points found in each BO round, and the right plot shows the highest performance of all the points found so far. As we can see, BO consistently selects points with better average performance in each round than random search, which is expected. However, for the highest performance results, BO tends to fall behind Random in the initial few rounds. This might be because our batch expected improvement heuristic aims to take advantage of the currently most promising regions by selecting most points of the batch in the same region (exploitation), while Random more evenly explores the entire space (exploration). Nevertheless, BO seems to quickly catch up after a few rounds and shows long-term advantages.

## M.3 More D-VAE experiments

D-VAE leverages the proposed asynchronous message passing in both its encoder and decoder. To understand deeper how the asynchronous message passing helps, we add some ablation studies of D-VAE on our neural architecture datasets. Firstly, we replace the asynchronous message passing in D-VAE with simultaneous message passing to construct a D-VAE (SMP) baseline. Secondly, we keep the D-VAE encoder unchanged, and replace the decoder with S-VAE's string-based decoder. The decoder now is a simple RNN with $\mathcal{O}(n)$ complexity instead of the original $\mathcal{O}(n^2)$ complexity, thus is much faster. We name this variant D-VAE (FAST). We compare these two variants with the original D-VAE and S-VAE on our 6-layer neural architectures. The results are shown in Table 3. We can see that D-VAE still in general has the best performance. The D-VAE (SMP) baseline shows inferior reconstruction accuracy due to its nonzero training loss caused by the simultaneous message passing. The D-VAE (FAST) shows similar generative ability to D-VAE. In terms of latent space predictive ability, it is inferior to D-VAE but better than S-VAE. This indicates that, it is beneficial to

Table 3: Generative ability and latent space predictive ability of D-VAE and its variants.

| Methods | Generative ability (%) | | | | Predictive ability | |
|---|---|---|---|---|---|---|
| | Accuracy | Validity | Uniqueness | Novelty | RMSE | Pearson's $r$ |
| D-VAE (SMP) | 92.35 | 99.75 | **65.98** | **100.00** | 0.455±0.002 | 0.885±0.001 |
| D-VAE (FAST) | **99.98** | **100.00** | 40.53 | **100.00** | 0.419±0.006 | 0.905±0.001 |
| D-VAE | **99.96** | **100.00** | 37.26 | **100.00** | **0.384±0.002** | **0.920±0.001** |
| S-VAE | **99.98** | **100.00** | 37.03 | 99.99 | 0.478±0.002 | 0.873±0.001 |

use asynchronous message passing in both D-VAE's encoder and decoder, rather than only using it to encode DAGs.

Nevertheless, using D-VAE (FAST) allows us to work on deeper neural architectures with much less training time due to its linear decoding complexity. Thus, we repeat our NAS experiments on 12-layer neural architectures. Our final found network has an error rate of 3.88%, comparable to many state-of-the-art NAS results in the macro space such as [33]. We plot our final 12-layer neural architecture in Figure 12.

Figure 12: Visualization of the final 12-layer neural architecture found by D-VAE (FAST).

## M.4 More visualization results for neural architectures

We randomly pick a neural architecture and use its encoded mean as the starting point. We then generate two random orthogonal directions, and move in the combination of these two directions from the starting point to render a 2-D visualization of the decoded architectures in Figure 13.

## M.5 More visualization results for Bayesian networks

We similarly show the 2-D visualization of decoded Bayesian networks in Figure 14. Both D-VAE and S-VAE show smooth latent spaces.

Figure 13: 2-D visualization of decoded neural architectures. Left: D-VAE. Right: S-VAE.

Figure 14: 2-D visualization of decoded Bayesian networks. Left: D-VAE. Right: S-VAE.