[Reviews · NeurIPS 2019]

Reviewer 1



The paper is well written -- I have a pleasant read of the paper. A few comments: 1. Theorem 1 is not very useful. The input to A is either a set or an ordered set. Only when it is an ordered one, it needs global indices to decide the order, right? If the input is a set without order, there is no need to refer to global indices. 2. The paper embeds computation graphs, not graph structure or function. The paper can do a better job of relating and differentiating the three concepts (graph structure, computation graph, function). Graph structure => computation graph => function The arrow => is an onto relation: the same graph structure indicates the same computation graph and so on. But the reverse is not true. Computation graph is defined on generic operations while functions need instantiated operations. Then your discussion of equivalence of computation graph might be easier: if any operations make two derived functions not equivalent, then the computation graph is not equivalent. 3. The title should be VAE for "Computation Graphs" not for "Directed Acyclic Graphs" because the paper does not embed graph structures. 4. Figure 7: why does the right plot much darker than the left plot in general -- the distribution of the BIC scores (pixel colors) should be the same, right? 5. I don't know the work NAONet very well, but it seems to me that it also embeds computation graphs to vectors. Why not include a fair comparison of the two?

Reviewer 2



Originality: The main architectural pieces (VAE, message passing) have been proposed previously, but the specific focus on directed graphs using these components is new to my knowledge. Quality: The proposal was technically sound, but the proposal has undesirable properties which were not addressed by the authors, and lacks sufficient empirical evaluation: 1. Does the proposal easily (from a practical implementation perspective) allow for batching? 2. Decoder sequence length. Instead of N steps with an RNN, the proposal's decoder uses N*(1+2+...+N-1) steps. This may limit the proposal to small graphs (the authors have only evaluated on a fixed, small graph size). 3. Lacks simple baselines, such as an RNN/LSTM (since this is claimed as a special case of the proposed framework, it would be nice to see that performance improves over the special case). 4. Lacks a strong graph-NN baseline. GCN has strong assumptions compared to a general message passing neural network. A concrete suggestion would be comparing to [Li et al 2018], which generates a graph using a sequence of add-node and add-edge decisions using a message passing network to represent the partial graph. Since Li et al deals with general graphs, showing an improvement over their model may indicate that the proposed restriction to DAGs is beneficial. 5. Ablation studies. Crucially, we do not know whether the proposed asynchronous message passing actually helps, versus using standard message passing with the proposed architecture. 6. Simplified experimental settings: fixed-size, small graphs. Clarity: The paper is generally written clearly, with a few exceptions: - The term "asynchronous message passing" is a bit confusing, since each node has to wait for its predecessors' message to be computed (hence it seems synchronous); the authors might consider defining asynchronous and synchronous precisely in the context of their proposal. - (Minor): Line 29 - awkward phrasing ("the answer is yes"). Significance: Due to the issues above the significance is low-medium. ----- Edit after author feedback, raising from 5 to 6 ---- The authors provided a comparison with [Li et al], ablations of the message passing, and evaluation on a larger NAS setting and variable-length setting - thanks for the efforts in implementing and adding these! The proposed method does show improvements over [Li et al] and the message passing ablation strengthens the case for the scheme proposed here (though performance drops significantly with variable length sequences). Due to the new evaluations I'll raise the review by one point. Minor: LSTMs could be used via consistently linearizing the graphs, but with the [Li et al] baseline I don't think this is needed.

Reviewer 3



Originality: Generating DAGs is a very new and important problem. My main concern is about the novelty of the method. - Both the encoding and decoding procedures of the proposed work are highly related to Ref.1. However, no discussion or comparison is given. - The asynchronous message passing algorithm is a straightforward extension of the original algorithm. Quality: In general, the paper is of good technical quality. All claims are well supported by theoretical analysis and experimental results. But both the objective function and the training strategy are missing in the main manuscript and are over-simplified in the supplementary. Clarity: The paper is well organized and clearly written. Ref.1: Yujia Li, et. al. Learning deep generative models of graphs. ICML, 2019.

[Author Response · NeurIPS 2019]

We thank all the reviewers for their insightful comments!

**R1**: **(1)** Regarding Theorem 1, yes, global indices are only needed in ordered cases. We add this to emphasize that for
unordered/unindexed cases, isomorphic DAGs will be encoded the same. **(2)** The onto relation from graph structure to
computation to function is indeed a nice and clear way to differentiate them; thank you. We will try to differentiate these
concepts better. **(3)** We will try to improve the title. **(4)** The darker plot might be because the two principal components
on the right explain less variance of training data than those on left. Thus, along the two principal components on the
right we will see less points from the training distribution. These out-of-distribution points tend to decode to not very
good Bayes nets, thus are darker. We validated this guess by checking the variance explained, which are 59% (left)
and 17% (right). This also indicates that our model learns a more compact latent space. Thank you for raising this
question. We will add this possible explanation in the revised version. **(5)** NAONet is not a generative model, and uses
task-specific grammars to encode only neural architectures. This paper focuses equally on DAG generation and DAG
optimization. We will consider a fair comparison in the future when particularly applying our model to NAS.

**R2**: For the points in "quality": **(1)** Our proposal supports batching. We have used a batch size of 32 and 128 in the
experiments. The implementation is not hard; please refer to the submitted code for details. **(2)** The $\mathcal{O}(N^2)$ decoding
steps is basically a design choice, rather than a limitation of the model. For example, one can make it $\mathcal{O}(N)$ by
predicting all edges of a node at the same time. We choose the current decoding scheme because it can model the
dependence between edges, but will discuss its possible simplifications in the revised version. **(3)** RNN/LSTM is not
applicable to DAGs. In 3.3, we state RNN is a special case of our model only when DAG is reduced to a chain of nodes.
That said, we did include the GraphRNN baseline which uses RNNs to generate rows of adjacency matrix. **(4)** Thanks
for suggesting the baseline DeepGMG from [Li et al 2018]. We agree it is beneficial to show D-VAE's advantages over
DeepGMG in modeling DAGs. As we cannot find the official code of DeepGMG, we strictly followed the paper to
implement it ourselves. Several modifications are made to adapt it to our tasks. First, we make it a VAE by equipping it
with a 3-layer message passing network as the encoder (using its own MP functions). Second, we feed in nodes using a
topo-order instead of the original random order (and see much improvement). Third, the sampled edges only point to
new nodes to ensure acyclicity. Then, we trained DeepGMG on our 6-layer NN dataset. We did a lot of hyperparameter
tuning, but the training loss never reached near zero. In comparison, D-VAE can be perfectly trained to near zero loss.
This results in DeepGMG's worse reconstruction accuracy (Table 1). This nonzero loss also acts like an early stopping
regularizer, making DeepGMG generate more unique graphs. Note that in our tasks, reconstruction accuracy is much
more important than uniqueness, since we need embeddings to perfectly remap to their original structures after latent
space optimization. Further, the predictive ability of DeepGMG embeddings is also worse, indicating it is less suitable
to perform optimization in its latent space.

**(5)** Thanks for suggesting the ablation study.
We replace D-VAE's asynchronous message
passing with Simultaneous Message Passing
to make the baseline "D-VAE (SMP)". This
model also has nonzero training loss, similar

| Methods | Generative ability (%) | | | | Predictive ability | |
|---|---|---|---|---|---|---|
| | Accuracy | Validity | Uniqueness | Novelty | RMSE | Pearson's $r$ |
| D-VAE | **99.96** | **100.00** | 37.26 | **100.00** | **0.384±0.002** | **0.920±0.001** |
| DeepGMG [Li et al 2018] | 94.98 | 98.66 | 46.37 | 99.93 | 0.433±0.002 | 0.897±0.001 |
| D-VAE (SMP) | 92.35 | 99.75 | **65.98** | **100.00** | 0.455±0.002 | 0.885±0.001 |
| D-VAE on 12-layer nets | 95.23 | 99.88 | 90.34 | 100.00 | 0.488±0.001 | 0.875±0.001 |
| D-VAE on mixed data | 70.45 | 90.76 | 77.12 | 100.00 | - | - |

to DeepGMG. Thus, the uniqueness is higher but the reconstruction accuracy is lower (Table 1). Regarding latent
space predictivity, it is worse than D-VAE and DeepGMG. **(6)** Regarding small graphs, we added one experiment
that trains our model on 20,000 12-layer neural networks. It achieves similarly good performance (Table 1). The best
12-layer network found after Bayesian optimization achieves a CIFAR-10 test error of 3.85%, comparable to many
state-of-the-art NAS results in macro space. We cannot really test D-VAE on NNs with hundreds or thousands layers,
since such datasets are hardly available. However, due to the combinatorial search space complexity, people also do not
search very deep neural architectures, but build deep ones by searching shallow cells and stacking them multiple times.
We leave this to future work. To show that our model is not limited to fixed-size graphs, we also train it on 20,000
graphs mixed of 6, 8, 10, 12-layer neural networks (5,000 each). The results are shown in Table 1's last row.
We will add all the above results into a revised version. Finally, we would like to respectfully argue that although our
proposal is inspired by many previous excellent works, it is not simply assembling them for a new problem. Instead, it
has made multiple customized innovations for DAGs where theoretical justifications are provided. For instance, the
injectivity w.r.t. computation (Theorem 1) ensures the two DAGs (representing the same computation) in main paper's
Figure 1 are encoded the same by asynchronous MP, where simultaneous MP will fail by encoding them differently.

**R3**: Thank you for acknowledging that generating DAGs is an important new problem to study! For the comparison
with DeepGMG [Li et al 2018], please refer to R2-(4) and Table 1. We will also add a discussion of the differences
between the two models. Basically, DeepGMG is not tailored for DAGs – there is no guarantee of acyclicity; DeepGMG
uses simultaneous message passing to encode graph structures, while D-VAE uses asynchronous message passing to
encode computations; after each decision step, DeepGMG requires multiple message passings for all nodes, while
D-VAE does one message passing only for the target node; and DeepGMG is not a VAE, thus does not have a latent
space for DAG optimization. We will add a thorough description of our training strategy in the main manuscript too.

[Meta-Review · NeurIPS 2019]

Although initial scores were mixed, after the rebuttal period reviewers all converged to acceptance. In this regard, some issues related to comparisons with existing work were adequately resolved. Beyond this, the demonstration of injectivity is also a nice analytical complement to the algorithmic and empirical contributions, and overall I enjoyed reading this work.